DATA RELEASE

# Potential vectors associated to Oropouche virus transmission in Cuba, 2024

Mónica Sánchez González[1,†], Ariamys Companioni[1,†], Eric Camacho[1], Silvia Serrano[1], Mayling Álvarez[1], Henry Rodriguez-Potrony[2], Yuneisy Alfonso[2], Barbara Liberty[3], Javier Varens[3], Yanet Martínez[1], Zulema Menendez[1], Dayana Rodríguez Velázquez[1], Madelaine Rivera[4], Daymi Lugo[4], Vivian Kouri[1], Maria G. Guzman[1] and Gladys Gutiérrez-Bugallo[1,*]

1 Center for Research, Diagnostic, and Reference, Institute of Tropical Medicine Pedro Kourí, PAHO-WHO Collaborating Center for the Study of Dengue and its Control, Havana, 17100, Cuba
2 Centro Provincial de Higiene, Epidemiologia y Microbiología, Santiago de Cuba, Cuba
3 Centro Provincial de Higiene, Epidemiologia y Microbiología, Cienfuegos, Cuba
4 Cuban Ministry of Health, Havana, Cuba

## ABSTRACT

From May to October 2024, Cuba experienced an outbreak of Oropouche virus (OROV), an Orthobunyavirus previously restricted to the Amazon region. As no Orthobunyavirus circulation had been previously reported in Cuba, the local vector involvement was uncertain. Entomo-virological surveys were conducted in active transmission areas across three provinces. Adult insects collected with traps and aspirators were screened for OROV by real-time RT-qPCR. A total of 2,180 specimens representing six dipteran species or families were identified. *Culex quinquefasciatus* and *Aedes aegypti* occurred in all provinces, with *Cx. quinquefasciatus* predominating (*n* = 1,785), followed by *Ae. aegypti* (*n* = 285) and Ceratopogonidae (*n* = 49). Eleven pools containing these taxa tested positive for OROV RNA. Detection of OROV in various species suggests possible involvement of multiple vectors in the Cuban outbreak. Further studies are needed to assess vector competence and elucidate OROV transmission dynamics in the Caribbean region.

**Subjects** Ecology, Biodiversity, Virology

**Submitted:** 19 August 2025

\* Corresponding author. E-mail: ggutierrezbugallo@gmail.com; gladysg@ipk.sld.cu

† Contributed equally.

Preprint submitted at https://doi.org/10.5281/zenodo.17665897

Included in the series: *Vectors of human disease* (https://doi.org/10.46471/GIGABYTE_SERIES_0002)

## INTRODUCTION

Oropouche virus (OROV) (order *Bunyavirales*, family *Peribunyaviridae*, genus *Orthobunyavirus*, Simbu serogroup) is an emerging arbovirus in South and Central America [1]. Initially detected in Trinidad and Tobago [2], OROV transmission has historically been confined to the Amazon Basin [3, 4], where it circulates primarily through a sylvatic cycle involving dipteran vectors, such as biting midges and mosquitoes, and vertebrate hosts, including sloths, non-human primates, and other mammals [5].

Due to its increasing geographic spread and potential for urban transmission, the Pan American Health Organization issued an alert in February 2024 highlighting the virus's potential to emerge beyond its traditional range [5]. This alert followed outbreaks reported in cities near the Amazon region, where human cases typically coincided with the rainy season, when vector populations increase [6].

The midge *Culicoides paraensis* (Goeldi, 1905) (Diptera: Ceratopogonidae) has been identified as the principal vector of OROV in various settings, including sylvatic, peri-urban, and urban outbreaks [3, 7, 8]. Among mosquitoes, *Culex quinquefasciatus* Say, 1823 has been suggested as a secondary vector due to its high abundance in endemic regions, such as Brazil and French Guyana [9, 10], and its limited but demonstrated vector competence in laboratory conditions [11–13]. Other species, such as *Coquillettidia venezuelensis* (Theobald, 1912) and *Aedes serratus* (Theobald, 1901), have also been proposed as potential vectors [2, 3].

The first documented outbreak of OROV beyond its typical geographic zone occurred in Cuba on May 27, 2024 [14]. Human cases of Oropouche fever were initially confirmed in Santiago de Cuba province, then in Cienfuegos and later in other parts of the country, totaling 506 confirmed cases by September 2024 [15]. Given the lack of prior Orthobunyavirus circulation on the island, knowledge about potential OROV vectors in Cuba is extremely limited.

In response, entomo-virological surveillance was implemented to identify insect species involved in the outbreak. Vector identification during an emerging arboviral event is essential for informing targeted control-measures and building effective response strategies [16]. Recognizing the vectors responsible for OROV transmission in Cuba is not only critical for national public health planning but also holds global relevance, as it represents the virus's first known establishment on a Caribbean island and its occurrence in both urban and rural environments.

Here, we present the results of the initial entomological investigations conducted in three Cuban provinces between May and October 2024.

## MATERIAL AND METHODS

### Collection sites and species identification

Insect collection was conducted at 14 active OROV transmission areas across three Cuban provinces between May and October 2024 (Figure 1): (1) Havana (Pulido Humaran, Grimau, and Puentes Grandes localities), (2) Cienfuegos (III and VIII localities), and (3) Santiago de Cuba (Armando García, Distrito José Martí, Caney, 30 de Noviembre, 28 de Septiembre, Finlay, Distrito Josué País, Julian Grimau, and Ernesto Che Guevara localities).

Specimens were collected using adult traps (BG-Sentinel traps with BG-Lure cartridges and New Jersey light traps), operated for 24 hours from 8:00 a.m. at each site. Adult insects were also collected with Prokopack aspirators at the same locations. Captures were conducted both indoors and outdoors, mainly in households with confirmed or suspected OROV cases.

Collection sites were categorized based on vegetation cover as follows: low (≤30%), moderate (30–70%), or high (≥70%) vegetation, following the criteria described by [17].

Collected specimens were stored at 4 °C during transportation and handling. Taxonomic identification was performed using established morphological keys for Culicidae [18] and Ceratopogonidae [19] at the Entomology Reference Laboratory of the Pedro Kourí Institute of Tropical Medicine.

Insects were sorted into pools of 5 to 25 individuals based on species, sex, collection date, and location. For female insects, only those that were visibly non-engorged were included in the pools. In addition, specimens in which the cold chain was not consistently maintained after collection were excluded from molecular analysis.



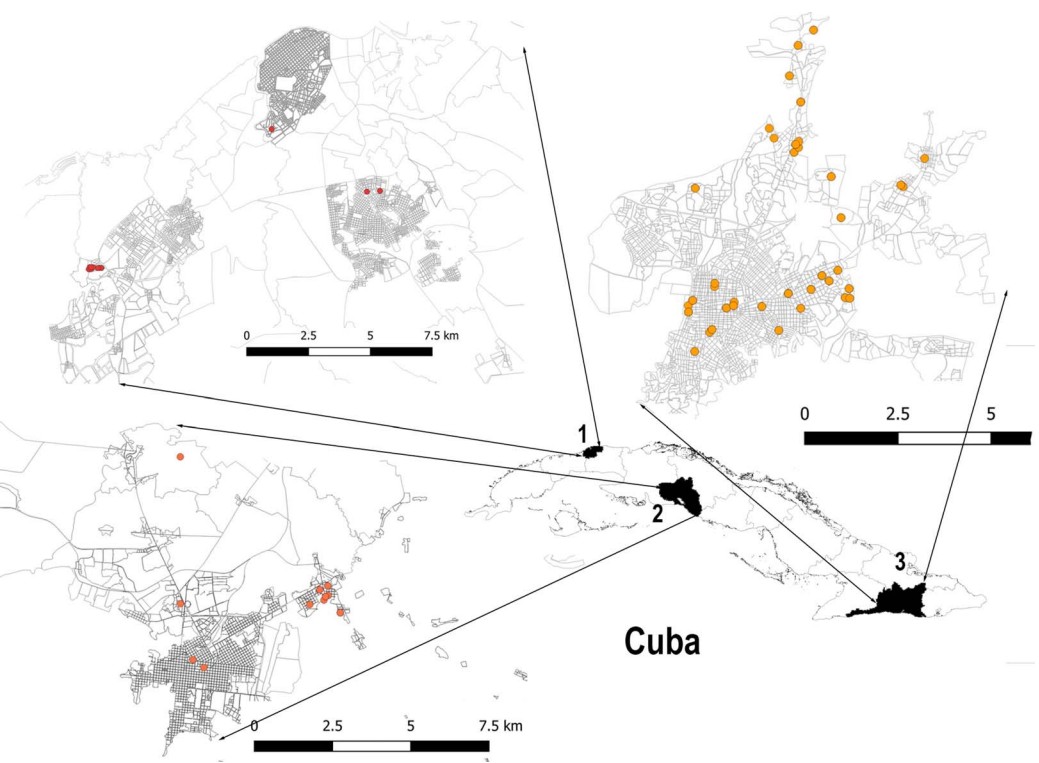

**Figure 1.** Locations of insect sampling (dots) using BG-Sentinel traps, New Jersey traps, and Prokopack aspirators in the provinces of Havana (1), Cienfuegos (2), and Santiago de Cuba (3), from May to October 2024.

## Nucleic acid extraction and PCR detection

Insect pools were homogenized in 500 μL of Dulbecco's Modified Eagle Medium (DMEM) supplemented with 10% fetal bovine serum. After centrifugation at 13,000×$g$ for 15 min at 4 °C, 140 μL of the supernatant was used for viral RNA extraction using the QIAamp Viral RNA Mini Kit (QIAGEN, Germany), following the manufacturer's instructions.

OROV RNA was detected by targeting a fragment of the S gene using a one-step real-time reverse transcription PCR (RT-qPCR) protocol, as described by [20]. Briefly, the reaction was carried out in a final volume of 20 μL using the SuperScript™ III Platinum® One-Step qRT-PCR System (Invitrogen, Thermo Fisher Scientific, USA), following the manufacturer's instructions. The primer pair 5′-TCCGGAGGCAGCATATGTG-3′ and 5′-ACAACACCAGCATTGAGCACTT-3′ was used at a final concentration of 0.3 μM each. The probe 5′-FAM-CATTTGAAGCTAGATACGG-3′ was used at a final concentration of 0.1 μM. The positive control consisted of a 63-base single-stranded DNA construct, and the negative control consisted of a pool of naïve *Ae. aegypti* mosquitoes reared at the Pedro Kourí Institute of Tropical Medicine insectary for over 30 generations. After RNA extraction, 10 μL of each sample was added to the reaction mixture.

The thermal cycling conditions were as follows: 50 °C for 15 min, 95 °C for 10 min, followed by 45 cycles of 95 °C for 15 s and 60 °C for 30 s. Samples were considered positive for OROV if the cycle threshold (Ct) value was less than 40.

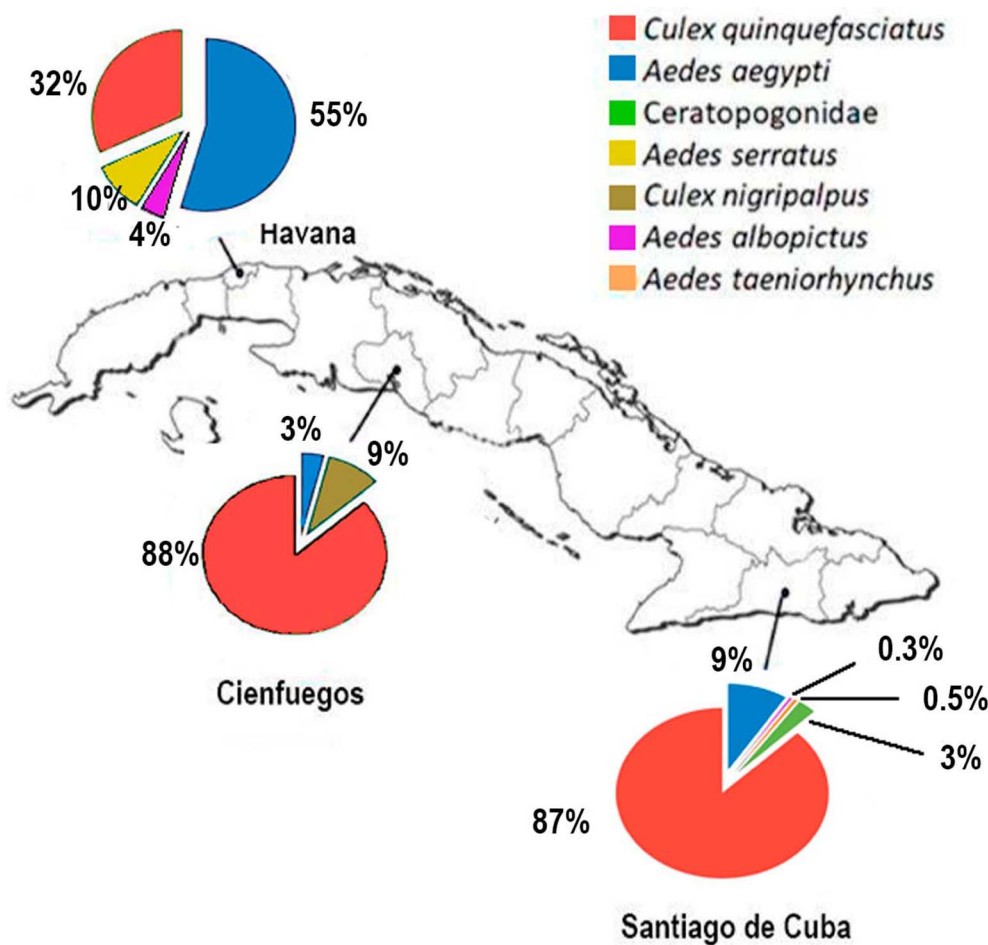

**Figure 2.** Species distribution by province (Havana, Cienfuegos, and Santiago de Cuba) during the collection period May–October 2024.

## Minimum infection rate estimation

The minimum infection rate (MIR) is an indicator of viral activity within a vector population [21]. MIR was calculated for each species using the following formula:

$$\text{MIR} = \frac{\text{positive pools}}{\text{total individual tested}} \times 1{,}000$$

## RESULTS

### Species collected in active OROV transmission sites

A total of 2,180 specimens were collected from 14 localities with active OROV transmission (Figure 2). These specimens belonged to six distinct species across two mosquito genera: *Aedes* (*Aedes aegypti* (Linnaeus, 1762), *Aedes albopictus* (Skuse, 1895), *Ae. serratus*, and *Aedes taeniorhynchus* (Wiedemann, 1821)) and *Culex* (*Cx. quinquefasciatus* and *Culex nigripalpus* Theobald, 1901), as well as members of the family Ceratopogonidae Newman, 1834.

Overall, the most frequently captured species was *Cx. quinquefasciatus* (*n* = 1,785), followed by *Ae. aegypti* (*n* = 285), Ceratopogonidae (*n* = 49), *Ae. serratus* (*n* = 21),



**Table 1.** Insect pools that were positive for Oropouche virus, collected from various localities in the Cuban provinces of Santiago de Cuba and Havana between May and October 2024.

| Species | N | Ct | Sex | Province | Locality | Collection date (d/m/y) | Collection method | Location |
|---|---|---|---|---|---|---|---|---|
| *Cx. quinquefasciatus* | 20 | 36 | F | Santiago de Cuba | Ernesto Che Guevara | 23/05/2024 | BG-Sentinel | Outdoors |
| *Cx. quinquefasciatus* | 20 | 39 | F | Santiago de Cuba | Ernesto Che Guevara | 24/05/2024 | BG-Sentinel | Outdoors |
| Ceratopogonidae spp. | 14 | 39 | F | Santiago de Cuba | Ernesto Che Guevara | 24/05/2024 | BG-Sentinel | Outdoors |
| *Cx. quinquefasciatus* | 20 | 39 | F | Santiago de Cuba | Ernesto Che Guevara | 26/05/2024 | BG-Sentinel | Outdoors |
| *Cx. quinquefasciatus* | 22 | 38 | F | Santiago de Cuba | Ernesto Che Guevara | 26/05/2024 | BG-Sentinel | Outdoors |
| *Cx. quinquefasciatus* | 24 | 27 | F | Santiago de Cuba | Armando García | 07/06/2024 | BG-Sentinel | Outdoors |
| *Cx. quinquefasciatus* | 25 | 21 | F | Santiago de Cuba | Armando García | 07/06/2024 | BG-Sentinel | Outdoors |
| *Cx. quinquefasciatus* | 25 | 39 | M | Santiago de Cuba | Armando García | 07/06/2024 | BG-Sentinel | Outdoors |
| *Ae. aegypti* | 12 | 37 | F | Santiago de Cuba | Finlay | 13/07/2024 | BG-Sentinel | Outdoors |
| *Ae. aegypti* | 13 | 38 | F | Havana | Puentes Grandes | 13/07/2024 | Prokopack | Indoors |
| *Ae. aegypti* | 8 | 39 | F | Havana | Grimau | 07/08/2024 | Prokopack | Indoors |

Ct: cycle threshold value from RT-qPCR; N: number of insects in the pool.

*Cx. nigripalpus* (*n* = 18), *Ae. albopictus* (*n* = 14), and *Ae. taeniorhynchus* (*n* = 8). Provincial distribution showed that *Cx. quinquefasciatus* accounted for the majority of captures in Santiago de Cuba (87%) and Cienfuegos (88%), while *Ae. aegypti* was the predominantly sampled species in Havana, comprising 55% of the total specimens collected there (Figure 2).

Overall, vegetation cover analysis at the sampled locations showed that the majority of captures occurred in areas with moderate vegetation cover. In Santiago de Cuba, *Cx. quinquefasciatus* was most frequently captured in areas with moderate vegetation cover (44%), followed by low (29%) and high (28%) cover. Ceratopogonidae were equally split between moderate cover (50%) and the combined high and low categories (25% each). *Ae. aegypti* occurred mainly in low vegetation areas (45%), with moderate (33%) and high (26%) cover less represented.

In Cienfuegos, captures for all species were highest in areas with moderate vegetation cover. *Cx. quinquefasciatus* was recorded at 52% in moderate, 33% in high, and 15% in low cover areas. *Cx. nigripalpus* occurred predominantly in moderate cover (60%), with the remainder in high cover (40%). All *Ae. aegypti* specimens were found in moderate cover areas.

In Havana, most captures occurred in sites with high vegetation cover. The proportion of specimens collected in high vegetation cover ranged from 100% for *Ae. albopictus* and *Ae. serratus*, to 70% for *Ae. aegypti*, and 65% for *Cx. quinquefasciatus*.

## Detection of OROV in entomological samples

A total of 1,238 insects were processed by RT-qPCR, of which 1,213 were apparently non-engorged females and 25 were males. Specimens were grouped into 81 female pools and one male pool, distributed as follows: 57 pools of *Cx. quinquefasciatus*, 17 of *Ae. aegypti*, 2 of *Ae. albopictus*, 2 of Ceratopogonidae, 2 of *Cx. nigripalpus*, 1 of *Ae. serratus*, and 1 of *Ae. taeniorhynchus*. The single male pool consisted of *Culex quinquefasciatus*.

In total, 11 pools (14%) tested positive for OROV by RT-qPCR, nine from Santiago de Cuba and two from Havana. Among these, 64% were composed of *Cx. quinquefasciatus*, 27% of *Ae. aegypti*, and the remainder of Ceratopogonidae (Table 1). All positive pools corresponded to specimens collected from May to August.

The highest MIR was estimated for *Ae. aegypti* (30.3), followed by Ceratopogonidae (20.4), and *Cx. quinquefasciatus* (7.1).



## DISCUSSION

Various arthropod species have been implicated as potential vectors of OROV in the Americas, primarily based on their field abundance and, in some cases, on the detection of viral RNA during outbreaks [22–25]. In this study, we investigated the insect species potentially involved in OROV transmission during the 2024 outbreak in Cuba.

Our results showed that *Cx. quinquefasciatus* was the most frequently collected species during the active transmission period, especially in areas with moderate and high vegetation cover. This species has previously been associated with outbreaks of Oropouche fever during the rainy season [9, 26, 27]. Known as the "southern house mosquito", *Cx. quinquefasciatus* is widespread and highly abundant in Cuba throughout the year, breeding both indoors and outdoors in a wide range of artificial containers [28].

Although laboratory studies indicate limited vector competence, some have shown that *Cx. quinquefasciatus* can achieve infection, viral dissemination, and transmission under certain conditions [13, 29]. However, other research from Brazil reported that OROV replication is restricted by midgut barriers in this species, preventing successful transmission [12]. Recent ecological-niche modeling suggests that *Cx. quinquefasciatus*, despite being considered a secondary vector, may have a slightly higher vector potential for OROV in Cuba than *C. paraensis* [30].

Interestingly, we detected OROV RNA in a pool of male *Cx. quinquefasciatus* mosquitoes. Similar findings have been reported for males of *Cx. quinquefasciatus* and *Ae. aegypti* [31, 32], suggesting the possibility of vertical and/or venereal transmission. This transmission route has been documented for other arboviruses, including dengue and Zika in *Ae. aegypti* [33, 34], and West Nile in *Cx. quinquefasciatus* [35].

*Ae. aegypti* was the second most abundant species collected. This mosquito is extremely common in Cuba and is the primary vector for dengue and Zika viruses on the island [28, 36]. Notably, *Ae. aegypti* exhibited the highest MIR among the species tested, suggesting a potential role in OROV transmission in these settings.

Although *Ae. aegypti* has never been incriminated as a vector for OROV in the field, this may be due to its absence in previously affected rural areas. Laboratory findings on its competence for OROV are conflicting. Some studies report that *Ae. aegypti* cannot become infected or transmit the virus via saliva [37, 38]. In contrast, other studies demonstrate that the species is refractory to oral infection but susceptible to systemic infection through intrathoracic inoculation, supporting high viral loads [38].

It is well established that vector competence varies not only between species but also among populations of the same species across different geographic regions, influenced by both genetic and environmental factors [9]. Additionally, ecological traits such as vector density and host preference can offset lower intrinsic vector competence [39]. Further studies should assess the ability of *Ae. aegypti* populations in Cuba to acquire, disseminate, and transmit OROV. In the meantime, the detection of viral RNA in non-engorged females, along with the species' abundance and human proximity, underscores its potential role as a vector in the current outbreak.

We also detected OROV RNA in one pool of Ceratopogonidae specimens. Although species-level identification was not possible at the time of processing, later entomological surveys confirmed the presence of *C. paraensis* in Cuba [40]. The absence of *C. paraensis* in our trap collections may reflect methodological limitations, as this species is notoriously

difficult to capture with standard trapping techniques [40], rather than a true absence from transmission areas.

In our study, Ct values for OROV-positive pools ranged from 21 to 39. Other studies have reported Ct values between 25.1 and 35.3 in *Ae. serratus* [41], and between 34.7 and 37.0 in *Ae. albopictus* and *Cx. quinquefasciatus* [32]. Since RNA is a labile molecule that degrades easily, especially under suboptimal storage conditions, we suspect that high Ct values in our study may be partially due to interruptions in the cold chain.

Arbovirus infection rates in vector populations typically vary across time and space [42]. For comparison, MIRs for the dengue virus in *Ae. aegypti* are generally reported to be below 10 [33]. However, MIR values are influenced by multiple factors, including vector-pathogen dynamics, ecological conditions, and sampling strategies [43]. In the case of OROV, recent studies have reported MIRs ranging from 0.2 to 2.3 in *Cx. quinquefasciatus*, *Ae. albopictus*, and *Limatus durhamii* Theobald, 1901 [9, 32]. Interestingly, in some instances, confirmed human cases of OROV have been reported without the concurrent detection of infected vectors in the affected areas. This apparent discrepancy may be explained by factors such as the high abundance of potential vector species, which can reduce the likelihood of detecting infected individuals when infection prevalence is low, or by limitations in sampling sensitivity [10].

In conclusion, this study identified *Cx. quinquefasciatus*, *Ae. aegypti*, and members of the family Ceratopogonidae as potential vectors of OROV in the Cuban context, marking a notable expansion beyond the virus's previously recognized geographic range. Although laboratory confirmation of vector competence is still required, the detection of OROV RNA in these species offers important preliminary evidence to inform public health decision-making. Vector control programs should take these findings into account to prioritize targeted interventions and mitigate the risk of future outbreaks. Moreover, this work contributes to the growing body of knowledge on the ecology and transmission dynamics of OROV, an emerging arbovirus of increasing regional and global concern.

## DATA AVAILABILITY

The dataset described here is available on the GBIF repository [44].

## EDITOR'S NOTE

This paper is part of a series of Data Release articles working with GBIF and supported by TDR, the Special Program for Research and Training in Tropical Diseases, hosted at the World Health Organization [45].

## ABBREVIATIONS

Ct, cycle threshold; MIR, minimum infection rate; OROV, Oropouche virus; RT-qPCR, reverse transcription PCR.

## DECLARATIONS

### Ethics approval and consent to participate

The authors declare that ethical approval was not required for this type of research.

### Competing interests

The authors declare that they have no competing interests.

## Authors' contributions

AC, HRP, YA, BL, JV, and ZM collected the specimens; AC, EC, HRP, YA, BL, JV, YM, and ZM identified the specimens; MR and DL coordinated the sampling efforts; SS and MA conducted the OROV detection; DRV and MS performed the spatial analysis; MSG, AC, YM, and GGB analysed the data; VK and MGG supervised the study; and MSG, AC, and GGB wrote the manuscript. All authors revised and approved the final version.

## Funding

No external funding was used.

## Acknowledgements

We sincerely thank the National Program for Vector Surveillance and Control, along with its provincial and municipal branches, as well as the Provincial Centers for Hygiene, Epidemiology, and Microbiology in the provinces of Santiago de Cuba and Cienfuegos, for their support with mosquito sampling. We are also deeply grateful to the Entomo-Virological Laboratory Network of the Americas (RELEVA) and the Pan American Health Organization (PAHO) for providing reagents, and we warmly acknowledge Kiam Sang Kim Leiva for his valuable assistance with data management.

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
