## [Reviewer Report]

Indicate in the comments box below whether you are happy with the changes made or if the manuscript is unacceptable.Comments on revised manuscriptThe authors have adequately addressed the comments, and the current version substantially improves the clarity and scientific quality of the paper. I am satisfied with the revisions made to the manuscriptIndicate in the comments box below whether you are happy with the changes made or if the manuscript is unacceptable.Comments on revised manuscriptThe authors have adequately addressed the comments, and the current version substantially improves the clarity and scientific quality of the paper. I am satisfied with the revisions made to the manuscript

---

## [Editor Report]

Editor’s AssessmentEditors Assessment: In 2024 Cuba experienced a significant outbreak of Oropouche virus (OROV), an Orthobunyavirus that causes an influenza-like illness transmitted by insect bites, particularly from Culicoides midges. Prior to this event these outbreaks had mainly been confined to the Amazon, and no documented circulation of Orthobunyaviruses had been reported in Cuba. Leaving the role of local vectors in transmission largely unknown. To investigate potential vectors entomo-virological surveillance is crucial to identify the species involved in the outbreak. This paper is one of a series of Data Release papers in GigaByte supported by TDR and the WHO describing datasets hosted in GBIF to tackle these data gaps in vectors of human disease data. This paper presents the data from investigations at 14 active OROV transmission areas across three Cuban provinces between May and October 2024. Specimens were collected using BG-Sentinel traps and nucleic acid based detection methods used to identify the species. A total of 2180 specimens were collected, and these pooled and processed by RT-PCR to look for virus. With Cx. quinquefasciatus being the most likely candidate based on the positive pools. And Ae. aegypti, and Ceratopogonidae spp. also flagged as potential vectors of OROV in the Cuban context. Peer review and data auditing found this data to be high quality. While more laboratory confirmation of vector competence is still required, this initial data and the detection of OROV RNA in these species offers important preliminary evidence to inform public health decision-making.Editor’s AssessmentEditors Assessment: In 2024 Cuba experienced a significant outbreak of Oropouche virus (OROV), an Orthobunyavirus that causes an influenza-like illness transmitted by insect bites, particularly from Culicoides midges. Prior to this event these outbreaks had mainly been confined to the Amazon, and no documented circulation of Orthobunyaviruses had been reported in Cuba. Leaving the role of local vectors in transmission largely unknown. To investigate potential vectors entomo-virological surveillance is crucial to identify the species involved in the outbreak. This paper is one of a series of Data Release papers in GigaByte supported by TDR and the WHO describing datasets hosted in GBIF to tackle these data gaps in vectors of human disease data. This paper presents the data from investigations at 14 active OROV transmission areas across three Cuban provinces between May and October 2024. Specimens were collected using BG-Sentinel traps and nucleic acid based detection methods used to identify the species. A total of 2180 specimens were collected, and these pooled and processed by RT-PCR to look for virus. With Cx. quinquefasciatus being the most likely candidate based on the positive pools. And Ae. aegypti, and Ceratopogonidae spp. also flagged as potential vectors of OROV in the Cuban context. Peer review and data auditing found this data to be high quality. While more laboratory confirmation of vector competence is still required, this initial data and the detection of OROV RNA in these species offers important preliminary evidence to inform public health decision-making.

---

## [Reviewer Report]

Upload additional filesDRR-202508-02-R01/stage_files/DRR-202508-02/Review MS/Data-Review-DRR-202508-02.docxReviewer name and names of any other individual's who aided in reviewer Chris HunterDo you understand and agree to our policy of having open and named reviews, and having your review included with the published papers. (If no, please inform the editor that you cannot review this manuscript.)YesIs the language of sufficient quality?YesPlease add additional comments on language quality to clarify if needed
Are all data available and do they match the descriptions in the paper? NoAdditional CommentsThe RT-qPCR results are summarised in the manuscript, but the full results tables should probably be shared. Also should those observations of viruses be included in the GBIF observations dataset? The pooling of samples is summarised in the manuscript, but its not related back to the observations data in GBIF so its hard to know exactly which individuals from which sampling event are in which pool.Are the data and metadata consistent with relevant minimum information or reporting standards? See GigaDB checklists for examples <a href="http://gigadb.org/site/guide" target="_blank">http://gigadb.org/site/guide</a>YesAdditional CommentsThe mosquito observation data meets GBIF /DwC metadata standards.Is the data acquisition clear, complete and methodologically sound?YesAdditional CommentsIs there sufficient detail in the methods and data-processing steps to allow reproduction?YesAdditional CommentsIs there sufficient data validation and statistical analyses of data quality? YesAdditional CommentsIs the validation suitable for this type of data?YesAdditional CommentsIs there sufficient information for others to reuse this dataset or integrate it with other data?NoAdditional Commentsit would be hard to integrate the GBIF observation data in any other study without the inclusion of which observations were positive/negative for OROV.Any Additional Overall Comments to the AuthorRecommendationMinor Revision

---

## [Reviewer Report]

Upload additional filesDRR-202508-02-R01/stage_files/DRR-202508-02/Review MS/gx-DR-1755626735_revCMD.pdfReviewer name and names of any other individual's who aided in reviewer Catalina Marceló-DíazDo you understand and agree to our policy of having open and named reviews, and having your review included with the published papers. (If no, please inform the editor that you cannot review this manuscript.)YesIs the language of sufficient quality?YesPlease add additional comments on language quality to clarify if needed
The use of language is appropriate.Are all data available and do they match the descriptions in the paper? YesAdditional CommentsAre the data and metadata consistent with relevant minimum information or reporting standards? See GigaDB checklists for examples <a href="http://gigadb.org/site/guide" target="_blank">http://gigadb.org/site/guide</a>YesAdditional CommentsIt is recommended that you include information about the project from which this research is derived. Additionally, include the OROV results in GBIF according to the Dw-Core vocabulary.Is the data acquisition clear, complete and methodologically sound?YesAdditional CommentsIt is recommended that the figures be improved, especially Figure 1.Is there sufficient detail in the methods and data-processing steps to allow reproduction?NoAdditional CommentsAll taxonomic keys that were used must be included. Please see the attachment belowIs there sufficient data validation and statistical analyses of data quality? NoAdditional CommentsConsidering the title of the Data Paper, ‘Possible vectors associated with Oropouche virus transmission in Cuba, 2024,’ and its scope, it is noteworthy that specimens of the Ceratopogonidae family (the most important when discussing OROV) are identified at the family level. While this does not limit the sharing of the dataset, it does limit the interpretation and analysis of the implications of virus transmission. This is evident when highlighting the role of secondary vectors such as Cx. quinquefasciatus and presenting the results for the Ceratopogonidae family in second place.Is the validation suitable for this type of data?YesAdditional CommentsHowever, I strongly recommend establishing limitations regarding the lack of identification, or completing the identification of specimens among the Ceratopogonidae family.Is there sufficient information for others to reuse this dataset or integrate it with other data?NoAdditional CommentsNot for Ceratopogonidae familyAny Additional Overall Comments to the AuthorRecommendationMajor Revision